# Ensuring confidentiality and privacy of cloud data using a non-deterministic cryptographic scheme

**John Kwao Dawson**[1]*, **Frimpong Twum**[2], **James Benjamin Hayfron Acquah**[2], **Yaw Marfo Missah**[2]

**1** Sunyani Technical University and Student at Kwame Nkrumah University of Science and Technology, Sunyani, Ghana, **2** Kwame Nkrumah University of Science and Technology, Kumasi, Ghana

* kwaodawson@stu.edu.gh

**Data Availability Statement:** All relevant data are within the manuscript and its Supporting Information files.

## Abstract

The amount of data generated by electronic systems through e-commerce, social networks, and data computation has risen. However, the security of data has always been a challenge. The problem is not with the quantity of data but how to secure the data by ensuring its confidentiality and privacy. Though there are several research on cloud data security, this study proposes a security scheme with the lowest execution time. The approach employs a non-linear time complexity to achieve data confidentiality and privacy. A symmetric algorithm dubbed the Non-Deterministic Cryptographic Scheme (NCS) is proposed to address the increased execution time of existing cryptographic schemes. NCS has linear time complexity with a low and unpredicted trend of execution times. It achieves confidentiality and privacy of data on the cloud by converting the plaintext into Ciphertext with a small number of iterations thereby decreasing the execution time but with high security. The algorithm is based on Good Prime Numbers, Linear Congruential Generator (LGC), Sliding Window Algorithm (SWA), and XOR gate. For the implementation in C#, thirty different execution times were performed and their average was taken. A comparative analysis of the NCS was performed against AES, DES, and RSA algorithms based on key sizes of 128kb, 256kb, and 512kb using the dataset from Kaggle. The results showed the proposed NCS execution times were lower in comparison to AES, which had better execution time than DES with RSA having the longest. Contrary, to existing knowledge that execution time is relative to data size, the results obtained from the experiment indicated otherwise for the proposed NCS algorithm. With data sizes of 128kb, 256kb, and 512kb, the execution times in milliseconds were 38, 711, and 378 respectively. This validates the NCS as a Non-Deterministic Cryptographic Algorithm. The study findings hence are in support of the argument that data size does not determine the execution time of a cryptographic algorithm but rather the size of the security key.

**Funding:** The author(s) received no specific funding for this work.

**Competing interests:** The authors have declared that no competing interests exist.

## 1. Introduction

The increased activities of humans have made communication complex. This has increased the need to secure these activities by securing the communication channels [1]. National Institute of Standards and Technology defines cloud computing as a ubiquitous enabling model based on pay-as-you-go services that allow the sharing of pooled resources and is considered one of the fastest-growing technology in the world [2]. This has become possible because of its associated advantages of larger geographical treatment, low capital expenditure, scalability, and the ability to access its services anywhere using the internet [3]. Outsourcing of data to cloud service providers makes them prone to attacks due to the unsafe channel for the transfer of data from the cloud client to the cloud service provider's server [4]. Data confidentiality and privacy play a major role in data security. In [5], a model was proposed to provide a suitable scheme that can help reduce the security challenges of privacy and confidentiality in the cloud as shown in Fig 1.

### 1.1 Cloud service models

Four service models are considered under cloud computing. Software-as-a-Service (SaaS) is a cloud-based computerized innovation that offers ubiquitous access to a web-based service over the Internet on a pay-as-you-use basis to cloud clients [6]. Software is deployed to be used by all entities who are clients of the service provider [7]. Platform-as-a-Service (PaaS), makes available platforms that allow the development of the applications as well as maintenance of the applications [8]. The cloud clients can create, plan, improve and assess the developed applications directly from the cloud and also monitor the development cycle of the applications. Infrastructure-as-a-Service (IaaS) serves as the basis upon which all other cloud services are built. This replaces the traditional data centers in the normal network architecture. The cloud service providers use the IaaS model to provide the infrastructure upon which cloud client can store their resources [9]. A Container-as-a-Service (CaaS) is where developers use a package for their entire programming task. The container contains all the coding needs, run timing, and configuring together with the libraries the system needs to execute a host machine [10].

### 1.2 Data confidentiality

Data confidentiality is the guarding of data from unauthorized persons and ensuring that their content is kept as secured as possible [11]. (See Fig 2). It is considered one of the basic security

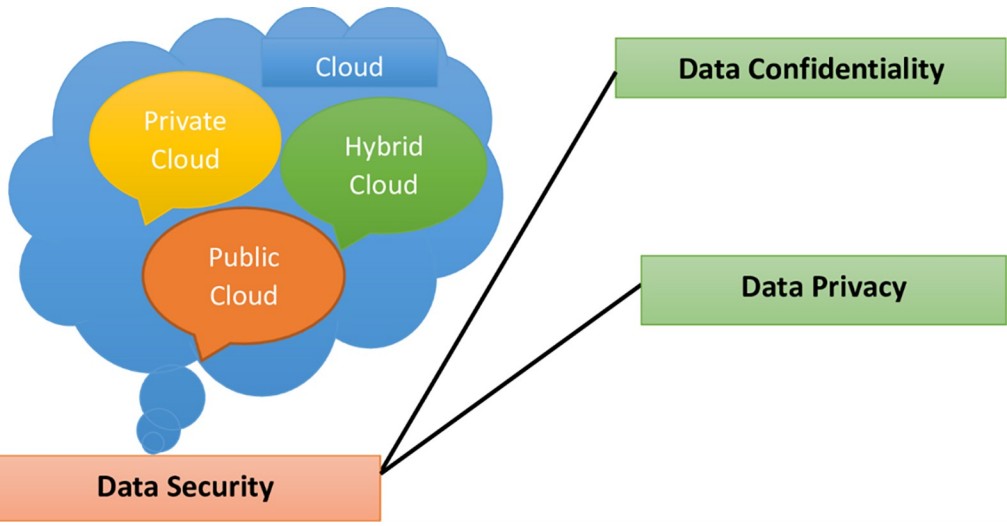

**Fig 1. Data security elements in cloud computing [5].**

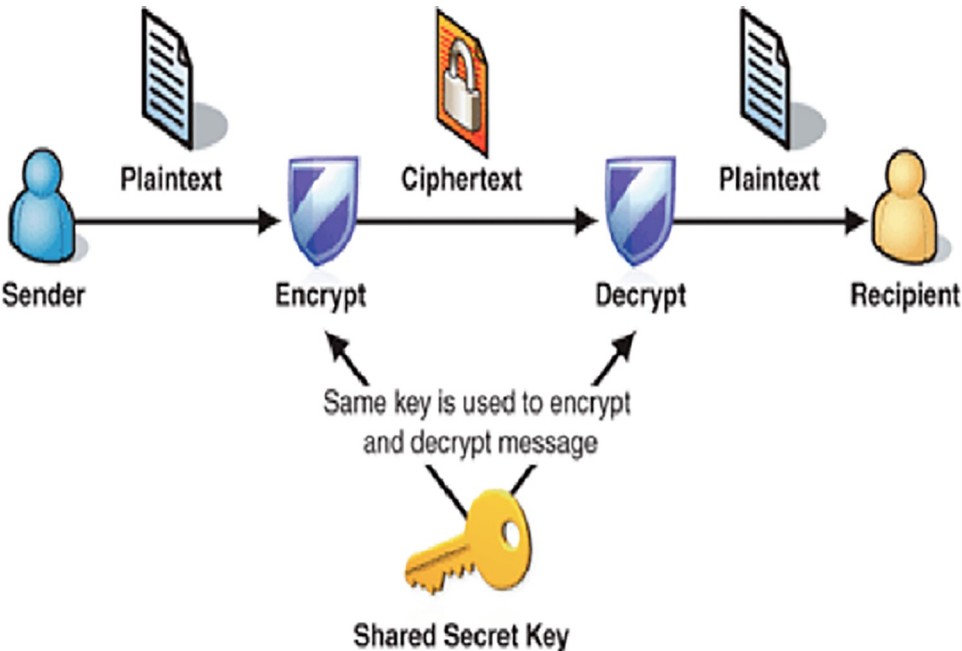

**Fig 2. A confidential scheme using a symmetric key** [3].

requirements for protecting data. This is vital in cloud computing due to the advantages of cloud computing such as offsite storage, multiuser schemes, and third-party utility providers [3]. Data confidentiality as a cloud security challenge occurs due to current security challenges resulting from flaws in systems design and its implementation [3] and [12].

### 1.3 Data privacy

Data Privacy is also considered information privacy, which considers the appropriate storage or handling of vital or sensitive information. Data privacy is the result of achieving confidentiality. The approach to ensuring data privacy in the cloud can be attained through the use of data security mechanisms shown in Fig 3 [13].

### 1.4 Problem statement

Researchers over the years have proposed several methods and techniques to address data confidentiality and privacy problems resulting from high execution times and predicted execution times based on data size [14–16]. However, the problem of confidentiality and privacy of data on the cloud with high and projected execution time persists [5,17,18]. Therefore, this paper proposes a new cloud data security scheme that integrates Good Prime numbers, Linear Congruential Generator (LCG), Sliding Window algorithm, and XOR gate. The approach achieves high security with few iterations and also enhances the execution times and makes it unpredictable.

## 2. Literature review

Several research has proposed varying cryptographic schemes aimed at ensuring cloud data confidentiality and privacy. Amongst them is the work suggested by Huang et al. [14]. In their work, they proposed an i-OBJECT scheme to ensure data confidentiality that depended on the fragmentation, decomposition of information, and the spread of divided data to distorted

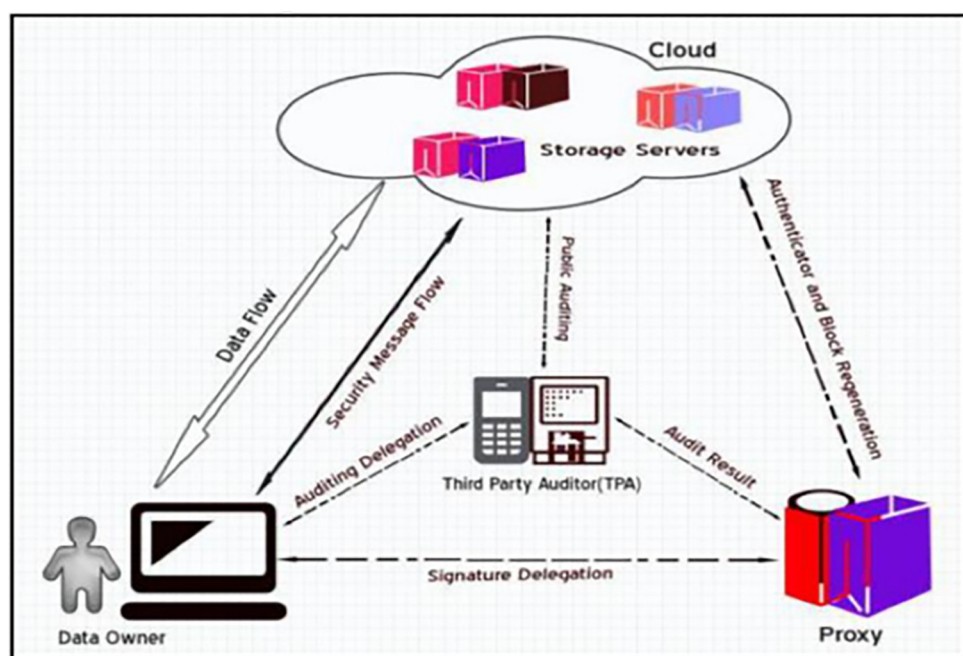

**Fig 3. Privacy structure in the cloud [13].**

cloud storage units. Their approach however did not thoroughly investigate the security against different configurations in the cloud which gave their system high security but with increased execution time. On the other hand, A Hard Decisional Composite Residuosity Assumption scheme which is an enhanced function of the Pailler encryption algorithm was proposed by El Makkaoui et al. [19] to ensure the confidentiality of data on the cloud. Their proposed algorithm's execution time was high. Jain and Kumar proposed a homomorphic cryptographic scheme to boost customers' conviction regarding the confidentiality of data. Their system allowed for data updates even in the encrypted form without the need for a security key from the cloud service provider. Their system resulted in a high execution time [20].

The work of Zhang et al. [21] proposed the use of a cryptographic scheme using a pairing-based algorithm based on blockchain that generates records that can resist tampering with records of patients to attain data privacy. Their system allowed all auditors on the system to verify the validity of the records but their contents were encapsulated. On the other hand, their approach failed to consider the security of e-health records under a cloud-assisted project and also depicted a high execution time in the data processing stages. Zhang et al. [15] again proposed an attribute-based access control scheme that is decentralized to achieve data confidentiality on the cloud. Their scheme helped to ensure repudiation which allowed for the generation of a secret key without an idea from the users of the system. However, due to the non-uniqueness of the attribute key, unauthorized users can decipher plaintext which increased the execution time as a result of complicity, which has a serious effect on the security of data.

Huang et al in achieving the same objective as Zhang et al. [22] proposed the use of a Lagrange interpolation-based control system to achieve data confidentiality of patient records on the cloud. Their system achieved this through the use of an authority-based scheme to access health records which increased the struggle in breaking the security of the database and accessing et health information. However, their approach had compatibility of systems and management of access problems as a result employed a lot of iteration which increased its execution time.

Rizwan et al. [23] proposed the use of Modular Encryption Standards (MES) integrated with the augmentation of condition-centric risk monitoring aiming to achieve confidentiality of health records. The confidentiality of data was attained by providing layered architecture of the health records. Making any decision regarding risk strategies of the MES is aided by a machine learning algorithm grounded using a Fuzzy Inference System integrated with Neural Networks. This system provides security against insider and outsider attacks by providing five variant keys for encryption. Their system however was not tested on other data types like image, audio, and video. Again there was proportionality between the data size and the execution time when textual data files were used.

Jain et al. [24] proposed the use of Secured Map Reduce to ensure the privacy of data on the cloud by introducing a layered interface between *Hadoop Distributed File System as well as Map-Reduce Layer. Their architecture provided privacy, solved expansion concerns in privacy, and ensured data mining tradeoff based on privacy utility but the iteration of the processes influenced the execution time negatively.*

Al-Balasmeh et al. [25] also ensured data privacy and information over vehicular cloud networks (VCNs) through the use of the data and location privacy (DLP) framework which secured the anonymity of personal data by providing location aided by obfuscation technique. In their work, much concentration was not given to securing loaded geo-fence storage infrastructure because it required many iterations to execute the process which has a negative influence on execution time.

Shivashankar and Mary ensured data privacy and reliability through the use of an enhanced Rider Optimization Algorithm (ROA) called Randomized Rider Optimization Algorithm (RROA). This framework used data sanitization and data restoration. The sanitization of data encapsulates the data from unauthorized users while data restoration is meant for data recovery. As a result of the number of iterations involved in data sanitization and restoration, execution time became proportional to the size of the data [26].

Hasan and Agrawal also proposed a new algorithm to ensure data privacy and confidentiality which was based on a probabilistic cryptographic scheme. Their approach used a single key which made it symmetric. Multiple encrypted data was able to represent plaintext which made it unrealistic in associating Ciphertext with plaintext. Despite the security strength of their proposed algorithm, it was still exponential. Their relation between data size and execution time was also proportional [27].

Gajmal and Udayakumar proposed a blockchain-based algorithm to ensure privacy as well as the utility of health data. The data privacy was achieved through the application of Tracy-Singh product aided by Conditional Autoregressive numbers at risk (CAViar)—based Bird Swarm scheme. This was used as an integration of BSA and CAViar to generate the privacy-preserving units. Their algorithm was effective but not efficient because the privacy percentage indicated a linear relationship between data size and time complexity [28].

The work of Shen et al. [29] proposed a proxy re-encryption scheme and oblivious random access memory that support the sharing of data on the cloud for data sharing. The encrypted data resulting from the implementation of the proposed algorithm allows the group to access and save the data resulting in the security of data sharing. Xu et al. [30] proposed a certificate-less auditing algorithm aimed at securing of sharing data and privacy of medical records. The execution times of the proposed algorithm are lower as compared with other state-of-the-art algorithms but had higher security and are also appropriate for sharing data on clouds.

In summary, the methodologies reviewed ensured confidentiality and privacy of data on the cloud. Despite ensuring data privacy and confidentiality in the cloud to maximize the benefits associated with the use of the cloud, existing schemes do not provide the resilience necessary against hackers. Their execution times were higher and were also proportional to the sizes of the data used in the execution process. This, therefore, requires a more robust scheme to

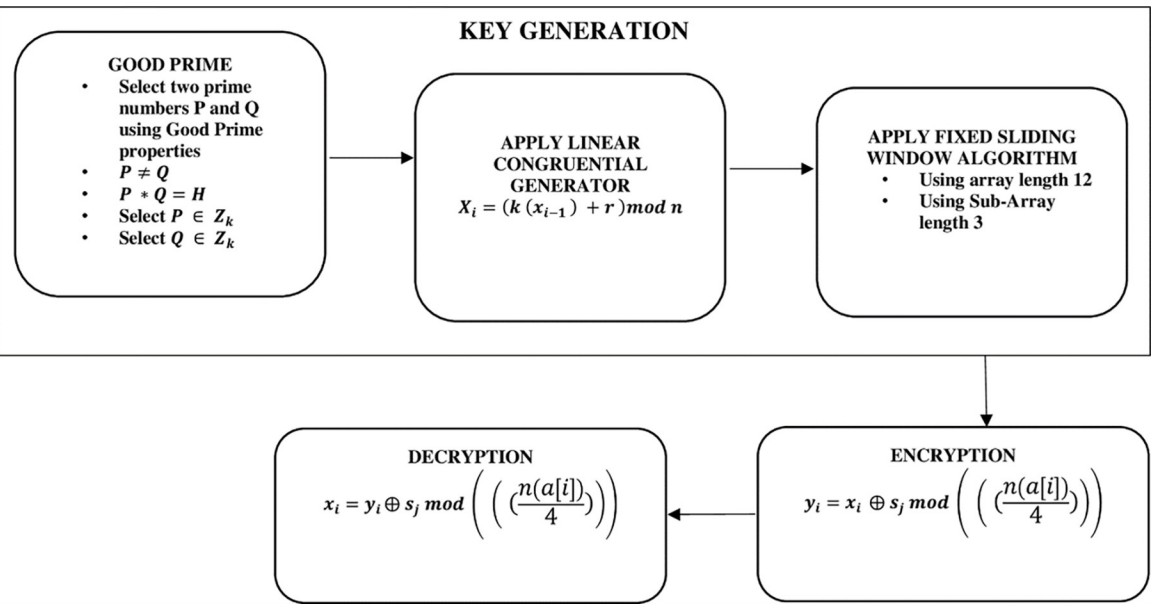

**Fig 4. Workflow diagram of proposed NCS algorithm.**

secure data against intruders with low and unpredicted execution times. This present study integrates Good Prime Numbers, Linear Congruential Generator, Fixed Sliding Door Window algorithm, and XOR circuit gate to frame a Non-Deterministic Cryptographic Scheme (NCS) to attain cloud data privacy and confidentiality with low and unpredicted execution time.

## 3. Methodology

A Non-Deterministic Cryptographic Scheme (NCS) is proposed to ensure cloud data privacy and confidentiality. The NCS is an integration of Good Prime, Linear Congruential Generator (LCG) [31], Fixed Sliding Window Algorithm (SWA), and XOR logic gate. The scheme is made up of three stages, including key generation, encryption, and decryption. The flow diagram demonstrating the proposed algorithm with the specified three stages is shown in Fig 4.

In this algorithm, two good prime numbers are selected at random. The product of the two good prime numbers is used as the seed value for the LCG. The other variables for the computation of the LCG are obtained based on Eqs 6, 7, 8 and 9 in this methodology. Twelve numbers are randomly selected from the numbers generated from the LCG. A fixed sliding window with sub-array three is applied to the twelve numbers selected. The plaintext is encrypted by computing $s_j$ (maximum value) [seen in Eq 7], modulus of the sub-array $\left(\frac{n(a[i])}{4}\right) \bigoplus x_i$ (the ASCII values of the alphabets). The decryption process is computed by calculating the modulus of $s_j$ (maximum values) and the sub-array $\left(\frac{n(a[i])}{4}\right) \bigoplus y_i$.

### 3.1 Key generation

**3.1.1 Good prime.** These are prime numbers whose squares are bigger than the product of two prime numbers in the sequence of primes at the same position before and after them [32]. The good prime numbers are generated using Eq 1.

$$P_n^2 > P(n-i) * P(n+i) \tag{1}$$

where $n$ is the list of prime numbers, $P(n-i)$ is the preceding prime number from the selected prime and $P(n+i)$ is the subsequent prime number, such that $1 \leq i \leq n-1$. As an example, the list of the first five prime numbers is; 2, 3, 5, 7, *and* 11. The first good prime from the first prime numbers will be computed using Eq 1 as; $5^2 > 3^*7, 5^2 > 2^*11$. From this, 5 can be considered a good prime. Therefore the first eight good prime numbers are [5, 11, 17, 29, 37, 41, 53, and 59]. Based on Eq (1), we select two random numbers $P$ and $Q$, such that $P^*Q = H$, $P \neq Q$, $P \in Z_k$, and $Q \in Z_k$. The resultant $H$ serves as the seed value for the Linear Congruential Generator which serves as the next stage in the key generation process.

**3.1.2 Apply linear congruential generator.** Any mathematical formula that results in the generation of categorization of randomized numbers computed based on a sporadic equation is considered Linear Congruential Generator (LCG) [31,33],. In generating the sequence of the values between $X1$, $X2$... and 0, $m-1$, the recursive relation of the numbers is shown by Eq (2).

$$X_{i+1} = (a\,X_i + C) mod\ m \tag{2}$$

Eq 2, work on the conditions such that; $m>0$, $a<m$, $c<m$, $X_o<m$

where $a$ = multiplier, $c$ = increment, $m$ = modulus, and $X_i$ = seed value. Using the product of $P$ *and* $Q$ as the seed value for Eq (2) generates a hundred thousand random numbers (100,000). The twelve numbers selected are the computational values for the Fixed Sliding Window Algorithm to generate the maximum and minimum values.

**3.1.3 Apply fixed sliding window algorithm.** The sliding window algorithm is used when identifying the results for a range of numbers in an array. This has the objective of converting a nested group of loops into a single loop which helps to reduce the complexity time from $O(n^2)$ to $O(n)$. The theory behind the sliding window algorithm is to generate the maximum or minimum units by computing the results continually for a range based on an array given [34].

Apply the Fixed Sliding Window algorithm using Eqs 3, 4, 5 and 6 to generate four ($a_y$, $a_{y1}$, $a_{y2}$, $a_{y3}$) numbers using a sub-array of 3 from the twelve number array depicted in Tables 1 and 2.

$$a_y = a_i + a_{i+1} + a_{i+2} \tag{3}$$

$$a_{y1} = a_{i+3} + a_{i+4} + a_{i+5} \tag{4}$$

$$a_{y2} = a_{i+6} + a_{i+7} + a_{i+8} \tag{5}$$

$$a_{y3} = a_{i+9} + a_{i+10} + a_{i+11} \tag{6}$$

**Table 1. Sliding window with 12 array length.**

| 2 | 6 | 4 | 8 | 3 | 2 | 7 | 9 | 11 | 13 | 15 | 1 |
|---|---|---|---|---|---|---|---|----|----|----|---|

**Table 2. Sliding window with 3 sub-array length.**

| 8 | 3 | 2 | 7 | 9 | 11 | 13 | 15 | 1 |
|---|---|---|---|---|----|----|----|---|

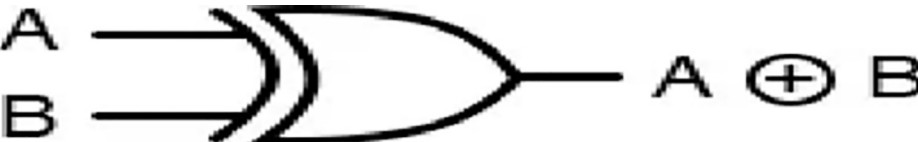

**Fig 5. $\bigoplus$ Gate** [35].

Use Eq 7 to select the maximum and minimum numbers from the four numbers computed. Where $s_j$ is the maximum value after applying the Fixed Sliding Window Algorithm on the 12 arrays generated?

$$s_j = \max(a_y,\, a_{y1},\, a_{y2},\, a_{y3}) \tag{7}$$

## 3.2 Encryption

The XOR gate ($\bigoplus$) shown in Fig 5, and Eq 8 is applied in the encryption process. In $\bigoplus$ gate, there is a combination of gates which results in the complex logic gate that is used more in constructing logic gates for arithmetic circuits, comparators for logic computation as well as detection of errors. In $\bigoplus$ gate, a "HIGH" voltage is obtained when all the input terminals are "DIFFERENT" and a "LOW" voltage when the input is all "HIGH" [35].

The $\bigoplus$ gate can be represented in Eq 8 as

$$A \bigoplus B = A\,B + A\,B \tag{8}$$

Encryption is the conversion of plaintext to Ciphertext and this is achieved by applying Eq 9 to the plaintext.

$$y_i = x_i \bigoplus s_j \bmod \left( \left( \frac{n(a[i])}{4} \right) \right) \tag{9}$$

The plaintext is encrypted by computing $s_j$ (maximum value) seen in Eq 7 modulus of the sub-array ($\frac{n(a[i])}{4}$). We then computed the results $\bigoplus x_i$ (the ASCII values of the alphabets) to obtain an eight-bit string for each alphabet.

## 3.3 Decryption

Decryption is the conversion of Ciphertext to plaintext and is computed based on the formula in Eq 10.

$$x_i = y_i \bigoplus s_j \bmod \left( \left( \frac{n(a[i])}{4} \right) \right) \tag{10}$$

The decryption process is computed by calculating the modulus of $s_j$ (maximum values) (see Eq 7) and the sub-array ($\frac{n(a[i])}{4}$). Compute the output $\bigoplus y_i$ to get an eight-bit string and the corresponding ASCII value is obtained to obtain the plaintext.

The proposed algorithm for the proposed scheme as indicated in Fig 4 is shown in Fig 6.

The architectural framework of the proposed algorithm is depicted in Fig 7. The plaintext for the cloud storage is converted to Ciphertext by applying Eq 9. The Ciphertext is then sent through the Internet Service provider's network for cloud storage. On request for the Ciphertext from the cloud service provider, Eq 10 is applied on the Ciphertext to convert it to plaintext for onward forwarding to the recipient.

Algorithm 1 Proposed Algorithm

**1:**   **Procedure NCS**
**2:**     Compute H = P *Q          $\rightarrow H : P, Q \in Good\ Prime$
**3:**   $X_Z = k\,(X_{x-1}) + r\ mod\ n$                  $\rightarrow (n > 0, 0 < k < n, 0 \le r < n, Compute\ the\ CLG$
**4:**   $(X_{x-1}) = H$
**5:**     $X_Z \in 1 \dots \dots \dots \dots 100,000$
**6:**     **For** $i = 0, i < 12, i++)$ **do**
**7:**         $\{a\,[i] = Rand\,(1, 100000)\}$
**8:**     **end for**
          *Apply Fixed Sliding Window (FSD) on 12 arrays susing sub − array of 3*
**9:**   $a_y = a_i + a_{i+1} + a_{i+2}$
**10:**  $a_{y1} = a_{i+3} + a_{i+4} + a_{i+5}$
**11:**  $a_{y2} = a_{i+6} + a_{i+7} + a_{i+8}$
**12:**  $a_{y3} = a_{i+9} + a_{i+10} + a_{i+11}$
**13:**  $s_j = \max(a_y, a_{y1}, a_{y2}, a_{y3})$
**14:**  $y_i = x_i \oplus s_i\ mod\ \left( \left( \frac{n(a[i])}{4} \right) \right)$                $\rightarrow ENCRYPTION$
**15:**  $x_i = y_i \oplus s_i\ mod\ \left( \left( \frac{n(a[i])}{4} \right) \right)$                $\rightarrow DECRYPTION$
**16:**  **End Procedure**

**Fig 6. The proposed NCS algorithm.**

## 4 Implementation, results, and discussion

### 4.1 Implementation

This section presents the implementation of the non-deterministic theory of the proposed NCS algorithm. The Sliding Window Algorithm (SWA) is applied to the twelve arrays of numbers generated from the application of LCG as shown in Fig 8. The maximum value selected which cannot be predicted under any circumstances gives the NCS algorithm the non-deterministic feature as shown in Fig 8. The message is then encoded by computing $s_j$ (maximum value) seen in Eq 7 modulus of the sub-array $(\frac{n(a[i])}{4})$. We then computed the results $\bigoplus x_i$ (the ASCII values of the alphabets) to obtain an eight-bit string for each alphabet as shown in Fig 9.

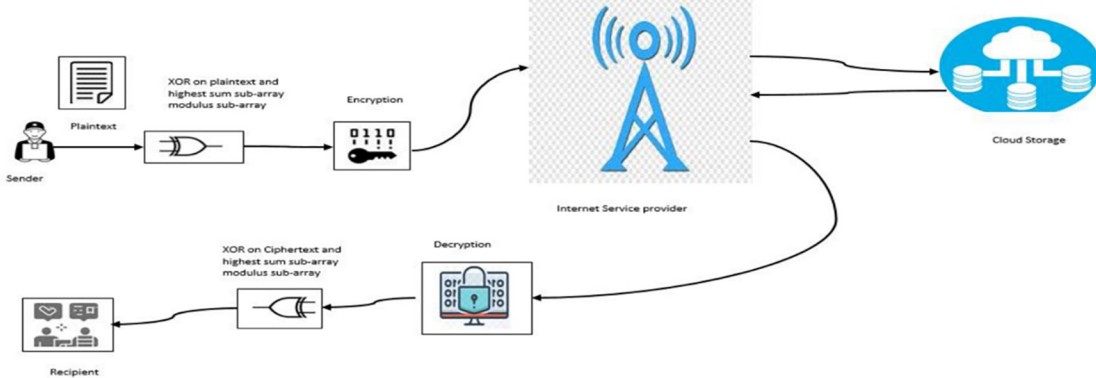

**Fig 7. Architecture for the proposed algorithm.**

```
int j = 0;
var sums = new List<int>();
while (j < numbers.Count) {
    var currentSum = 0;
    for (int i = j; i < j + 3; i++) {
    currentSum += numbers[i]; }
            j += 3;
    sums.Add(currentSum); }
    var max = sums.Max();
```

**Fig 8. Applying SWA to generate maximum sum in c#.**

The deciphering process is calculated by finding the modulus of $s_j$ (maximum values) (see Eq 7) and the sub-array ($\frac{n(a[j])}{4}$). The output is then $\bigoplus y_i$ (Ciphertext) to get an eight-bit string and the corresponding ASCII value is obtained to achieve the plaintext as shown in Fig 10.

**Data and environment for the experiment.** The experiment was conducted on an i7 Lenovo computer, a 2.10GHz CPU, and implemented using the C# language. Personal data was generated for the experimental work in Figs 11, 12, and 13. A predesigned dataset [36] was used to test the average execution times for NCS. The dataset was again used for the comparative analysis of the execution times for NCS, DES, AES, and RSA based on the data size used in the work of Ali et al. [37]. Personal data size of 4bk was executed using NCS and the corresponding Ciphertext and plaintext are shown in Figs 12 and 13.

```
var buffer = Encoding.ASCII.GetBytes(message);
var computedValues = new List<string>();
        var sb = new StringBuilder();
    for (int a = 0; a < buffer.Length; a++)
                    {
        var subArrMod = max % 3;
    var xor = buffer[a] ^ subArrMod;
    var y1 = Convert.ToString(xor, 2);
            sb.Append(y1);
    computedValues.Add(y1);
```

**Fig 9. Encryption of plaintext into eight-bit string in NCS.**

```
var decryptedBuffer = new List<byte>();
        for (int n = 0; n < computedValues.Count; n++)
        {
            var subArrMod = max % 3;
            var xor = Convert.ToInt32(computedValues[n], 2) ^ subArrMod;
            var x1 = Convert.ToString(xor, 2);
            decryptedBuffer.Add(Convert.ToByte(x1, 2));
        }
        Console.WriteLine("-----Decryption Time-------");

Console.WriteLine($"{TimeSpan.FromTicks(Stopwatch.GetTimestamp()).Mill
iseconds}ms");
        Console.WriteLine("-----Decryption Time-------");
        var plainText = Encoding.ASCII.GetString(decryptedBuffer.ToArray());
```

**Fig 10. Decryption of encoded data into plaintext using NCS algorithm.**

## 4.2 Results

The scheme was tested using a message size of $2^{n.}$ kb ($n \in 1, 3, 5, 7, 9$). The scheme was run 30 times to generate 30 different encryption and decryption times and their average was computed as shown in Tables 3 and 4 and Figs 14 and 15. The execution times for the comparative analysis for NCS, AES, DES and RSA are shown in Tables 5 and 6 and Figs 14 and 15.

## 4.3 Discussion

Time complexity refers to the time it takes for an algorithm to encipher a plaintext into a Ciphertext and decipher a Ciphertext into a plaintext. When the scrambling time is faster, this implies the data processing time is faster and such algorithms are considered to be efficient [38]. There have been a series of research to propose algorithms with the lowest execution time with a nonlinear time complexity $O(n)$ [39]. In this paper, a Non-Deterministic Cryptographic Scheme is proposed which is a symmetric algorithm. The algorithm was run 30 times to generate 30 different encryption and decryption times, and their averages were computed to consider the performance metrics of execution times of the algorithm using different data sizes. A comparative analysis was conducted based on the execution times of existing cryptographic algorithms such as RSA [37], AES [37], and, DES [37] as shown in Tables 5 and 6 and Figs 16 and 17.

From Tables 3 and 4, and Figs 12 and 13, it can be deduced that the average encryption time is high for a smaller data size of 2Kilobyte (KB) with an increased decryption time but

This refers to the time it takes for an algorithm to encipher a plaintext into a Ciphertext and decipher a Ciphertext into a plaintext. When the scrambling time is faster, this implies the data processing time is faster and such algorithms are considered to be efficient [26]. The comparative scrambling time for the proposed algorithm and classical algorithms is shown in figure 7. The proposed algorithm's (NCS) performance output obtained through its execution is compared with existing cryptographic algorithms such as Data Encryption Standard (DES), RSA, Advanced Encryption Standard (AES) as well as Message Digest -5 (MD5). Data Encryption Standard, Advanced Encryption Standard (AES), as well as the proposed algorithm Non-Deterministic Cryptographic Scheme (NCS), is considered to be symmetric algorithms while MD5, RSA are considered asymmetric schemes. The xi value is the ASCII value of the plaintext input from the user while si is the highest sum of the sub-array obtained from using the fixed sliding window, modulus the sub-array. During the encryption process, the value for the si and the modulus value using the sub-array is computed first after which the value is XOR with the ASCII values for the plaintext to obtain the Ciphertext in eight-bit string. A Non-Deterministic Cryptographic Scheme (NCS) is proposed to ensure cloud data privacy and confidentiality. A Non-Deterministic Cryptographic Scheme is an integration of Good Prime, Linear Congruential Generator (LCG), Fixed Sliding Window Algorithm (SWA), and XOR logic gate.This scheme is made up of three levels of key generation and encryption and decryption stages. The three levels of key generation are meant to produce secret keys which help to raise the security of the algorithm.Two good prime numbers at random. The product of the two good prime numbers is used as the seed value for the LCG. The other variables for the computation of the LCG are obtained based on equations 6, 7, 8, and 9. Twelve numbers are randomly selected from the numbers generated from the LCG. A fixed sliding window with sub-array three is applied to the twelve numbers selected. An encryption and decryption algorithm is then implemented using the plaintext and the highest value of the sub-array and using the sub-array as the modulus.his refers to the time it takes for an algorithm to encipher a plaintext into a Ciphertext and decipher a Ciphertext into a plaintext. When the scrambling time is faster, this implies the data processing time is faster and such algorithms are considered to be efficient [26]. The comparative scrambling time for the proposed algorithm and classical algorithms is shown in figure 7. The proposed algorithm's (NCS) performance output obtained through its execution is compared with existing cryptographic algorithms such as Data Encryption Standard (DES), RSA, Advanced Encryption Standard (AES) as well as Message Digest -5 (MD5). Data Encryption Standard, Advanced Encryption Standard (AES), as well as the proposed algorithm Non-Deterministic Cryptographic Scheme (NCS), is considered to be symmetric algorithms while MD5, RSA are considered asymmetric schemes. The xi value is the ASCII value for the letters of the plaintext input from the user while si is the highest sum of the sub-array obtained from using the fixed sliding window, modulus the sub-array. During the encryption process, the value for the si and the modulus value using the sub-array is computed first after which the value is XOR with the ASCII values for the plaintext to obtain the Ciphertext in eight-bit string.

**Fig 11. Plaintext to be encrypted with a file size of 4kb.**

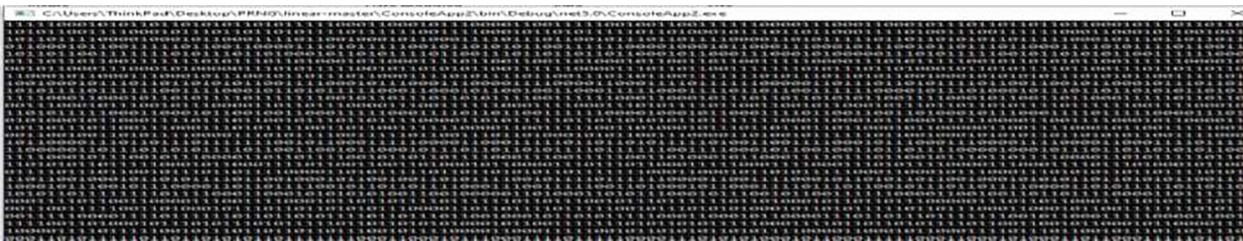

**Fig 12. Corresponding Ciphertext.**

reduced when the data size was increased to $2^3$KB with an increased decryption average time of 540.0667 Millisecond. There was also an observed reduction in average encryption time using a data size of $2^5$ KB with an increased average decryption time for the proposed algorithm. There was a tremendous reduction in average decryption time with an increased encryption time for the proposed algorithm when the data size was raised to $2^7$ KB. This is supported by the work of [40] that the execution time of an algorithm depends on the function of the size of the key used as the security key. The output of the averages of the proposed algorithm overrides the idea of the works of [37,41,42] indicating that file size is proportional to the encryption and decryption time of an algorithm.

Tables 5 and 6 and Figs 16 and 17 show the comparison between the encryption and decryption time of AES, DES, RSA, and the proposed algorithm NCS. From Table 5 and Fig 16, it can be observed that using a key size of 128 KB, NCS has the lowest encryption time followed by AES, DES, and RSA. This indicates the superiority of AES over other conventional algorithms but not against the proposed NCS algorithm [43]. Again using a key length of 512KB, the proposed algorithm has the lowest encryption time as opposed to RSA which has the highest. This confirms the theory that symmetric algorithms have the lowest encryption time during text file processing [38]. On the other hand, using a key size of 256KB, the decryption time for DES and RSA were the same while the proposed algorithm had the lowest. Contrarily, with a data size of 128kb, the execution times were lower but increased with a data size of 256kb and decreased again when the data size was increased to 512kb for the proposed NCS algorithm. It was justified that data size does not determine the execution time of an algorithm but is dependent on the function of the size of the security key [40].

### 4.4 Novelty

In an attempt by researchers to propose algorithms with low execution times, their time complexity remains linear [44]. Our proposed algorithm produces a nonlinear time complexity as depicted in Tables 5 and 6 and Figs 16 and 17. The execution times are dependent on the size of the security key and not the data size. For the proposed NCS algorithm, the encryption and

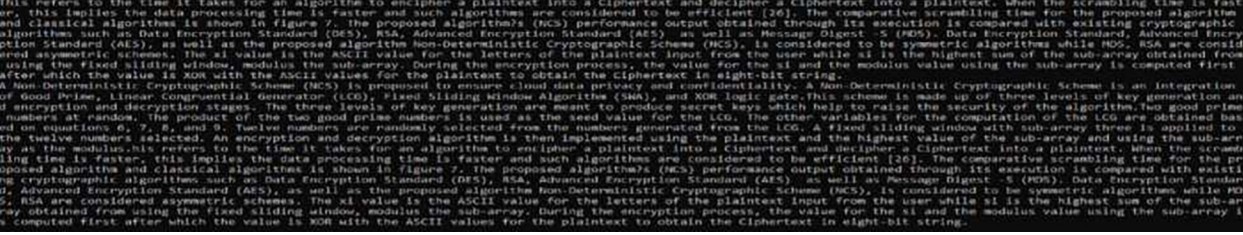

**Fig 13. Decrypted text.**

**Table 3. The proposed total encryption and average encryption time of NCS.**

| File Size | Total of 30 Encryption Time (ms) | Average Encryption Time(ms) |
|---|---|---|
| 2KB | 14415 | 480.5 |
| 8KB | 13147 | 438.2333 |
| 32KB | 10081 | 336.0333 |
| 128KB | 15426 | 514.2 |
| 512KB | 17558 | 585.2667 |

**Table 4. The proposed total decryption and average decryption time of NCS.**

| File Size | Total of 30 Decryption Time (ms) | Average Decryption Time(ms) |
|---|---|---|
| 2KB | 15726 | 524.2 |
| 8KB | 16202 | 540.0667 |
| 32KB | 16704 | 556.8 |
| 128KB | 14096 | 469.8667 |
| 512KB | 15877 | 529.2333 |

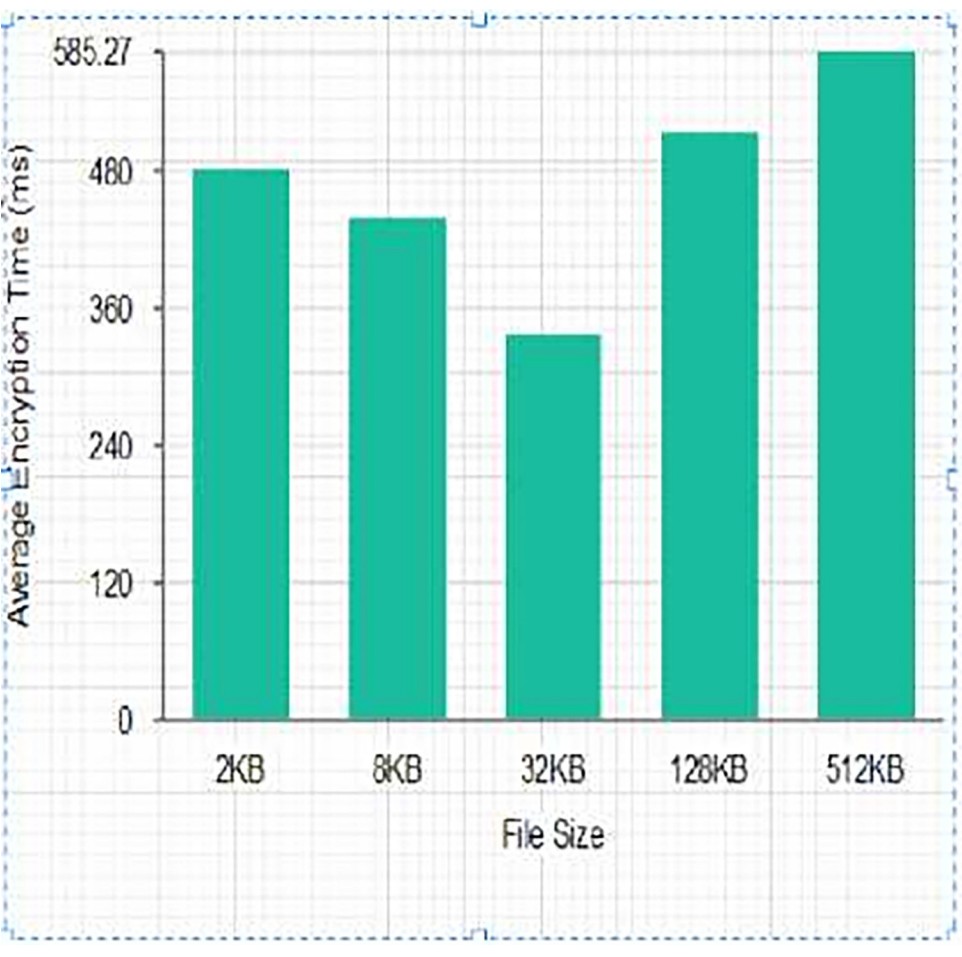

**Fig 14. Average Encryption Time (ms).**

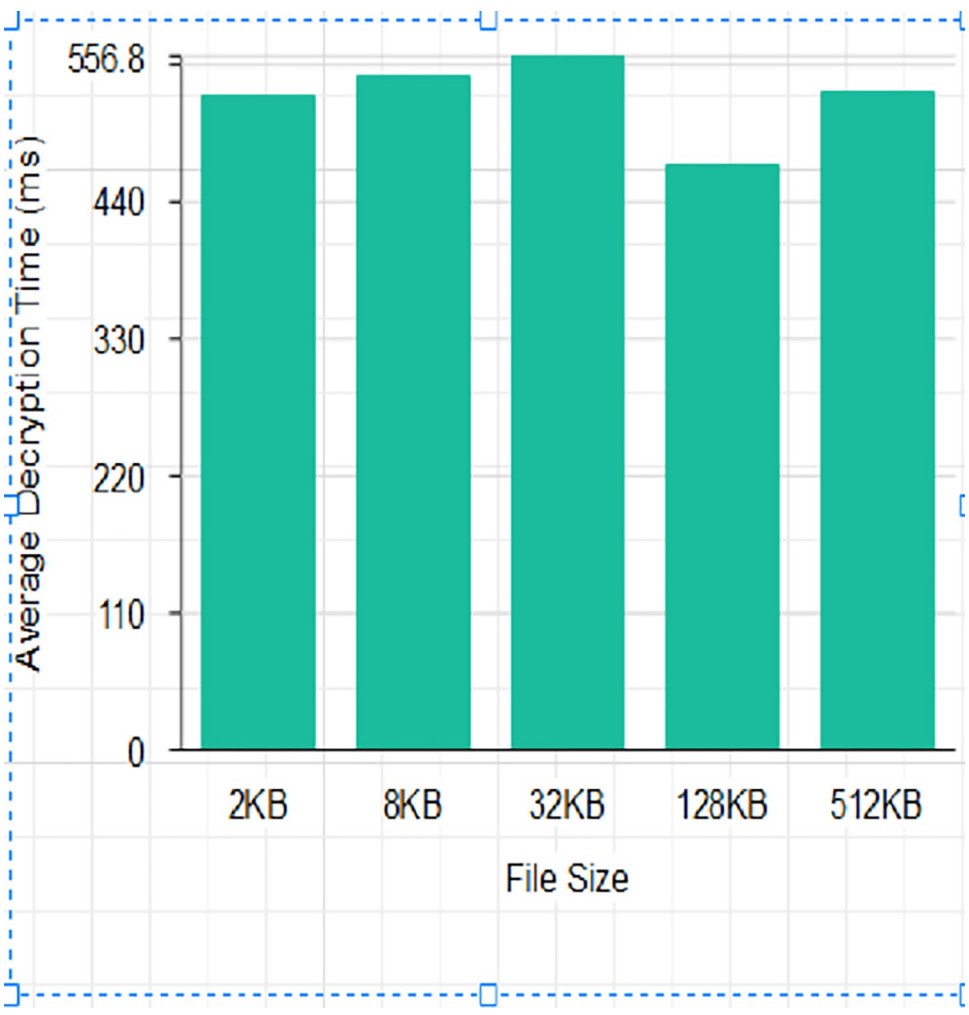

**Fig 15. Average Decryption Time (ms).**

**Table 5. Comparing encryption time of proposed (NCS) algorithm with conventional algorithms using different key sizes.**

| Algorithm | Encryption Time in Milliseconds | | |
|---|---|---|---|
| | **128kb** | **256kb** | **512kb** |
| AES | 2600 | 3500 | 4200 |
| DES | 3000 | 4100 | 5100 |
| RSA | 3300 | 4500 | 5400 |
| NCS | 192 | 552 | 358 |

**Table 6. Comparing the decryption Time of the proposed (NCS) algorithm with conventional algorithms using different key sizes.**

| Algorithm | Decryption Time in Milliseconds | | |
|---|---|---|---|
| | **128kb** | **256kb** | **512kb** |
| AES | 2600 | 3500 | 4200 |
| DES | 3000 | 4100 | 5100 |
| RSA | 3300 | 4500 | 5400 |
| NCS | 38 | 711 | 378 |

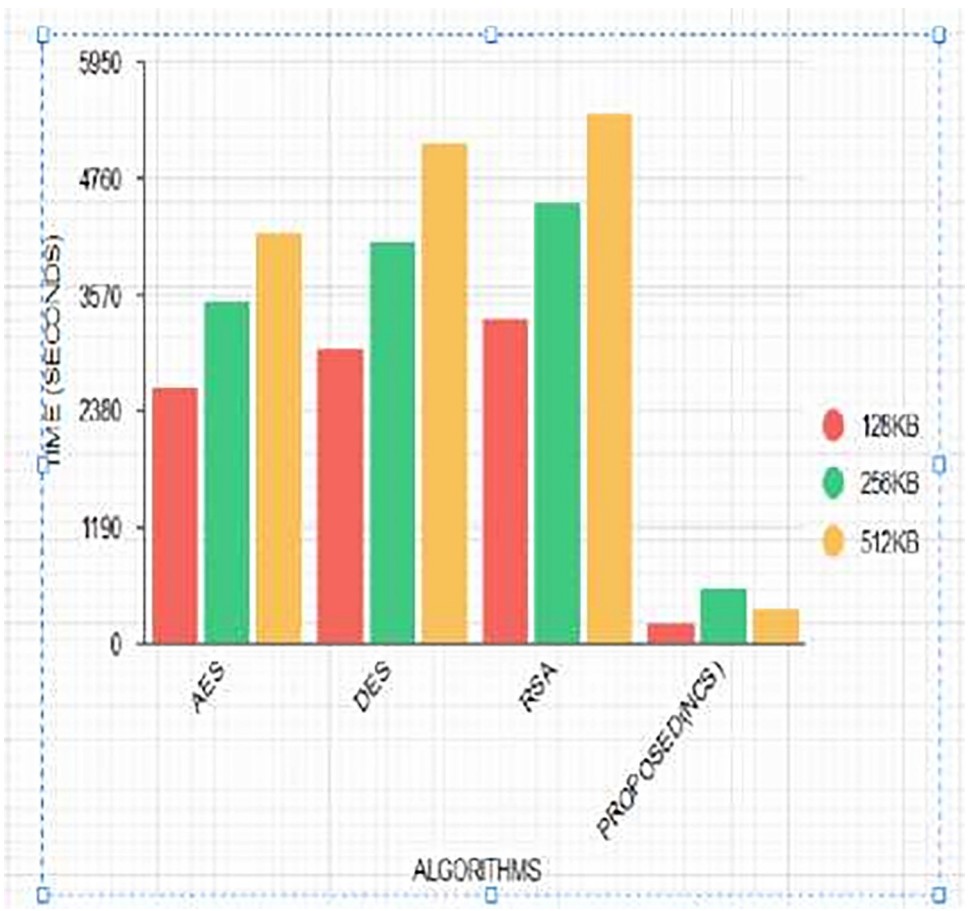

**Fig 16. Comparing encryption time of proposed (NCS) algorithm with conventional algorithms using different key sizes.**

decryption times are dependent on the value of $s_j$ as shown in Eq 7. This, therefore, influenced the execution times when data size was increased from 128kb, 256kb to 512kb with corresponding encryption times of 129ms, 552ms, and 358ms and decryption times of 38ms, 711ms, and 378ms. This makes our proposed NCS algorithm execution time nondeterministic.

## 5 Conclusion

The proposed algorithm was run 30 times using a dataset from Kaggle considering the data sizes used in the works of Ali et al. [37] and their averages computed to give execution times performance for the proposed NCS algorithm. The performance evaluation of three conventional algorithms (AES, DES, and RSA) together with the proposed algorithm was also evaluated based on their encryption and decryption times. There were differences between the encryption times for AES, DES, RSA, and NCS. The proposed NCS algorithm had the lowest encryption time with RSA having the highest. With decryption, with a data size of 256KB, the decryption time was higher compared with a data size of 512KB for the proposed NCS algorithm. From the results obtained, the security strength of the proposed algorithm (NCS) is stronger compared with industry-embraced algorithms like AES, DES, and RSA. Future works should be conducted to evaluate the memory consumption rate, CPU usage, and throughput against different file sizes.

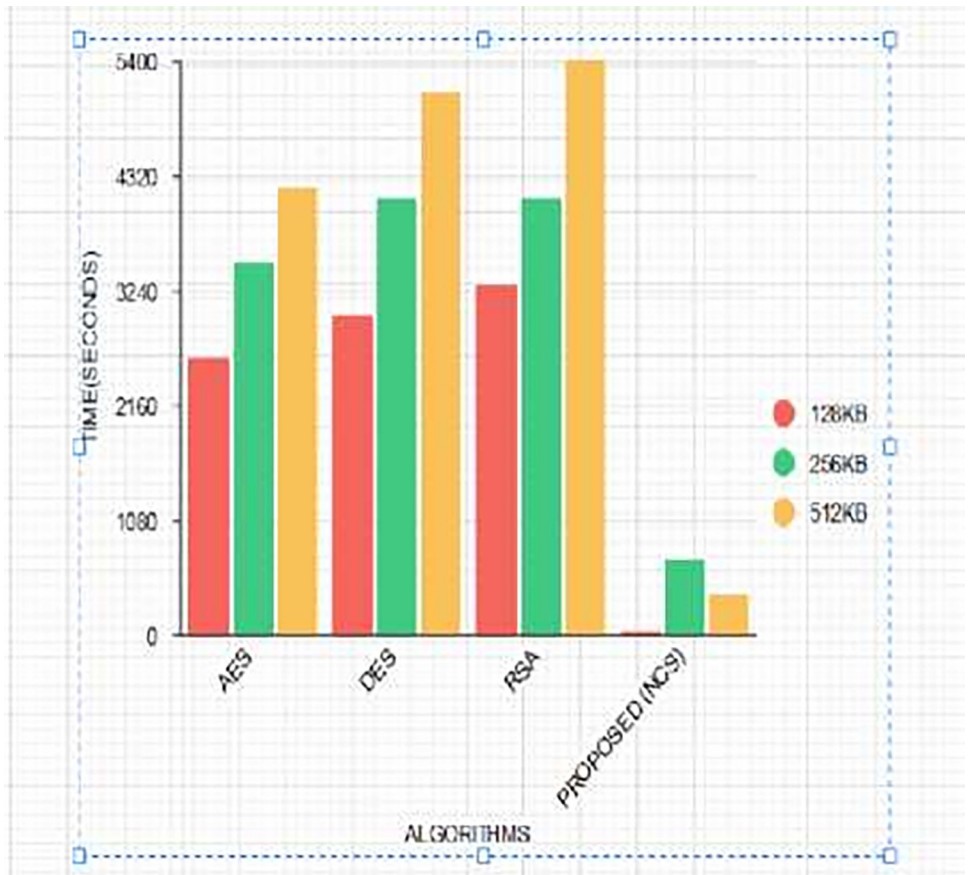

**Fig 17. Comparing the decryption Time of the proposed (NCS) algorithm with conventional algorithms using different key sizes.**

## Supporting information

**S1 Dataset.**
(TXT)

## Author Contributions

**Conceptualization:** John Kwao Dawson.

**Methodology:** John Kwao Dawson.

**Software:** John Kwao Dawson.

**Supervision:** Frimpong Twum, James Benjamin Hayfron Acquah, Yaw Marfo Missah.

**Validation:** John Kwao Dawson.

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
