## [Decision Letter · Decision Letter 0]

18 Apr 2022

PONE-D-22-10288ENSURING CONFIDENTIALITY AND PRIVACY OF CLOUD DATA USING A NON-DETERMINISTIC CRYPTOGRAPHIC SCHEMEPLOS ONE

Dear Dr. Dawson,

Thank you for submitting your manuscript to PLOS ONE. After careful consideration, we feel that it has merit but does not fully meet PLOS ONE’s publication criteria as it currently stands. Therefore, we invite you to submit a revised version of the manuscript that addresses the points raised during the review process.

We look forward to receiving your revised manuscript.

Kind regards,

Pandi Vijayakumar, Ph.D

Academic Editor

PLOS ONE

Journal Requirements:

3. Please include in your Data availability details of all the testing datasets used in this study, and describe int he methods section the origins of the data set  used, and how it was obtained for this study.

4. Please ensure that you refer to Figure 5 in your text as, if accepted, production will need this reference to link the reader to the figure.

5. We note you have included a table to which you do not refer in the text of your manuscript. Please ensure that you refer to Tables 1 and 2 in your text; if accepted, production will need this reference to link the reader to the Table.

Additional Editor Comments:

The paper has many serious issues which must be solved in the next round while submitting the revised version.

Reviewers' comments:

Reviewer's Responses to Questions

**Comments to the Author**

1. Is the manuscript technically sound, and do the data support the conclusions?

Reviewer #1: No

Reviewer #2: Yes

2. Has the statistical analysis been performed appropriately and rigorously? 

Reviewer #1: No

Reviewer #2: Yes

3. Have the authors made all data underlying the findings in their manuscript fully available?

Reviewer #1: No

Reviewer #2: Yes

4. Is the manuscript presented in an intelligible fashion and written in standard English?

Reviewer #1: No

Reviewer #2: Yes

5. Review Comments to the Author

Reviewer #1: Authors proposed a non-deterministic cryptographic scheme that integrates Good Prime Numbers, LCG, SWA, and

XOR Gate to achieve confidentiality and privacy of cloud data. Paper does not seem suitable in its current form. It should revised as per the comments given below.

* Improve the quality of provided figures.

* Add formal security analysis using the RoR model.

* Threat model should be in its proper form.

* Proposed scheme should be compared with some recently published schemes.

* Comparisons of communication cost, computation cost, storage cost, "security and functionality features" are not available.

Reviewer #2: This work proposes a new scheme for ensuring confidentiality and privacy of cloud data using a nondeterministic cryptographic scheme. After careful examination, I discovered numerous serious flaws in this paper.

1. There are many grammatical errors in the manuscript. So strong English proof reading is required. For example, “content is kept as secured as possible [16]” must be “content is kept as secure as possible [16]”. Also, check the statement “This helped to quicken the convolution and also helps in the analysis”

2. In page 14, what is the meaning of “Using Good Prime generate two prime numbers P and Q”. Also before this step, the authors should have used Select and select Q step which is used as the last step in Figure 4.

3. There are many formatting mistakes in the paper. For example, in page 15, after equation (2) a citation [27] has been used which should be given inside the text.

4. In the results and analysis section, the authors should do mathematical analysis between the proposed work with existing schemes. So table 3 and 4 should be modified.

5. Security analysis must be done using some formal verification tools or formal mathematical analysis.

6. Related work section is weak. Because Some important recent references are missing, the following references must be totally added in the Section "References" (otherwise, the reference is not enough, then it must be revised again until it is enough):

A privacy-preserving and untraceable group data sharing scheme in cloud computing

Mgpv: A novel and efficient scheme for secure data sharing among mobile users in the public cloud

A Flexible and Privacy-Preserving Collaborative Filtering Scheme in Cloud Computing for VANETs

Certificateless Public Auditing Scheme with Data Privacy and Dynamics in Group User Model of Cloud-Assisted Medical WSNs

Key management and key distribution for secure group communication in mobile and cloud network

6. PLOS authors have the option to publish the peer review history of their article (what does this mean?). If published, this will include your full peer review and any attached files.

Reviewer #1: No

Reviewer #2: No

---

## [Author Response · Author response to Decision Letter 0]

2 Aug 2022

COMMENT

Citations establish the credibility of research work and are poorly used

RESOLUTION/REBUTTAL

Enough citations have been added to the manuscript and are properly cited in the work. These can be found in the pages indicated

PAGES

4,5,6,15, 16

COMMENT

the literature review is shallow

RESOLUTION/REBUTTAL

The literature has been re-reviewed using articles that are directly linked to data privacy and confidentiality

PAGES

4,5,6

COMMENT

the novelty of the work in terms of comparison to existing comparable models is missing

RESOLUTION/REBUTTAL

A new section is included under the heading novelty and it indicates the contribution of this article to existing knowledge. Comparison is also done with existing models

PAGES

14,15, 17

COMMENT

the abstract states a number of facts that have not been cited, the abstract should in the brief state the novel contribution of the manuscript.

RESOLUTION/REBUTTAL

The abstract has been re-written indicating the problem statement, the novelty of the work

PAGES

1

COMMENT

source and citation of the dataset used are missing

RESOLUTION/REBUTTAL

The source of the dataset has been indicated as can be seen in the abstract and section 4.11

PAGES

1,13

COMMENT

equations have not been numbered uniquely and referred to in the text.

RESOLUTION/REBUTTAL

The equations have been uniquely numbered and cited in the text as can be found in section 3.1.1, 3.1.2

PAGES

8,9,10

---

## [Decision Letter · Decision Letter 1]

1 Sep 2022

ENSURING CONFIDENTIALITY AND PRIVACY OF CLOUD DATA USING A NON-DETERMINISTIC CRYPTOGRAPHIC SCHEME

ENSURING CONFIDENTIALITY AND PRIVACY OF CLOUD DATA USING A NON-DETERMINISTIC CRYPTOGRAPHIC SCHEME

PONE-D-22-10288R1

Dear Dr. Dawson,

We’re pleased to inform you that your manuscript has been judged scientifically suitable for publication and will be formally accepted for publication once it meets all outstanding technical requirements.

Kind regards,

Pandi Vijayakumar, Ph.D

Academic Editor

PLOS ONE

Additional Editor Comments (optional):

Reviewers' comments:

Reviewer's Responses to Questions

**Comments to the Author**

1. If the authors have adequately addressed your comments raised in a previous round of review and you feel that this manuscript is now acceptable for publication, you may indicate that here to bypass the “Comments to the Author” section, enter your conflict of interest statement in the “Confidential to Editor” section, and submit your "Accept" recommendation.

Reviewer #1: All comments have been addressed

Reviewer #2: All comments have been addressed

2. Is the manuscript technically sound, and do the data support the conclusions?

Reviewer #1: Yes

Reviewer #2: Yes

3. Has the statistical analysis been performed appropriately and rigorously? 

Reviewer #1: Yes

Reviewer #2: N/A

4. Have the authors made all data underlying the findings in their manuscript fully available?

Reviewer #1: Yes

Reviewer #2: Yes

5. Is the manuscript presented in an intelligible fashion and written in standard English?

Reviewer #1: Yes

Reviewer #2: Yes

6. Review Comments to the Author

Reviewer #1: Paper has been updated as per the given comments. The quality of the paper has been improved. Now I recommend acceptance of the paper

Reviewer #2: ALL THE COMMENTS GIVEN BY THE REVIEWERS ARE ADDRESSED.

HENCE THE MANUSCRIPT COULD BE ACCEPTED IN THE PRESENT FORM

7. PLOS authors have the option to publish the peer review history of their article (what does this mean?). If published, this will include your full peer review and any attached files.

Reviewer #1: No

Reviewer #2: No

---

## [Editor Report · Acceptance letter]

5 Sep 2022

PONE-D-22-10288R1 

Ensuring Confidentiality and Privacy of Cloud Data Using a Non-Deterministic Cryptographic Scheme 

Dear Dr. Dawson:

I'm pleased to inform you that your manuscript has been deemed suitable for publication in PLOS ONE. Congratulations! Your manuscript is now with our production department. 

Kind regards, 

on behalf of

Dr. Pandi Vijayakumar 

Academic Editor

PLOS ONE